# Protective Effects of Fish (Alaska Pollock) Protein Intake against Short-Term Memory Decline in Senescence-Accelerated Mice

**DOI:** 10.3390/nu14214618

**Published:** 2022-11-02

**Authors:** Yuki Murakami, Ryota Hosomi, Ayano Nishimoto, Toshimasa Nishiyama, Munehiro Yoshida, Kenji Fukunaga

**Affiliations:** 1Department of Hygiene and Public Health, Kansai Medical University, 2-5-1, Shinmachi, Osaka 573-1010, Japan; 2Faculty of Chemistry, Materials, and Bioengineering, Kansai University, 3-3-35, Yamate-cho, Osaka 564-8680, Japan

**Keywords:** fish protein, fish oil, senescence-accelerated mouse prone 10, cognitive decline, demyelination

## Abstract

Dietary fish intake has proven to have health benefits in humans. *n*-3 polyunsaturated fatty acids (PUFAs) in fish oil (FO), especially, may provide protection against age-related cognitive disorders. Owing to the unique benefits of *n*-3 PUFAs, other nutrients, such as fish protein (FP), have not been well studied. To clarify the effects of FO and FP on brain function, we investigated whether FO or FP feeding can prevent age-related cognitive dysfunction in senescence-accelerated mouse-prone 10 (SAMP10) mice. The FP group maintained a better working memory compared to the control and FO groups in the Y-maze test, but not episodic memory in the novel object recognition test. To evaluate demyelination levels, we measured neurofilament H (NfH) and myelin basic protein (MBP) immunoreactivity in the hippocampus (Hipp). Axon morphology was maintained in the FP group, but not in the control and FO groups. Additionally, the percentage of positive area for double-staining with NfH/MPB was significantly higher in the Hipp of FP-fed mice than in the control (*p* < 0.05). These results suggest that FP intake prevents age-related cognitive dysfunction by maintaining axonal morphology in the Hipp of SAMP10 mice.

## 1. Introduction

Dementia is a serious public health issue worldwide, particularly in developed countries. Previous studies have demonstrated that diet is one of the major factors associated with normal brain function. Especially, recent epidemiological studies have indicated that high dietary fish intake can significantly reduce the risk of dementia and mild cognitive impairment (MCI) [1,2]. Furthermore, a recent cross-sectional study elucidated that a higher frequency of fish intake reduced vascular brain damage before the onset of cerebrovascular diseases that may lead to the development and progression of cognitive impairment and dementia [3]. While fish intake can protect against cognitive decline, its effect on overall dementia continues to be debatable.

Dietary fish is commonly regarded as an important source of *n*-3 polyunsaturated fatty acids (PUFAs), which have proven therapeutic potential for dementia [4,5]. Several studies have investigated whether patients with Alzheimer’s disease (AD) benefit from supplementation with docosahexaenoic acid (DHA) and eicosapentaenoic acid (EPA), the main types of *n*-3 PUFAs [2,6,7]. In general, no convincing relationship was established between the use of *n*-3 PUFAs supplements and symptomatic improvement in patients with mild to moderate AD; further, the benefits of *n*-3 PUFAs appear to be inconsistent in research.

It is worth mentioning that fish is a rich source of protein. A previous in vitro study demonstrated that fish protein can interact with α-synuclein and inhibit the formation of amyloid plaques, which are primarily composed of fibrils formed by β-amyloid peptides in AD [8]. Our previous study also demonstrated that fish protein (FP) prepared from Alaska pollock attenuates liver steatosis and insulin sensitivity, and can improve physiological conditions [9]. Despite the beneficial role of FP in several diseases, its protective effects against dementia require further study.

The senescence-accelerated mouse (SAM) strain has been developed by phenotypic screening of spontaneous, premature aging, and short-lifespan littermates of the AKR/J stain. SAM-prone (SAMP)10 mice show not only a short lifespan but also age-related learning and memory deficits, mood disorders [10], and cognitive impairments [11], which are associated with a loss of synapses [12], a decrease in neuronal dendritic spine density, an abnormal blood-brain barrier, and atrophy in the brain prefrontal cortex [11]. Furthermore, the activity level of sphingomyelinase (SMase), an enzyme associated with demyelination in the brain [13], showed a significant increase in the cerebral cortex of SAMP10 mice in an age-dependent manner [14]. Myelin is an electrical insulator essential for action potential conduction and provision of trophic support to neuronal axons of the central nervous system (CNS) [15]. Loss of oligodendrocytes, maintenance of myelin, and the subsequent demyelination may trigger pathological events that may intercept regional CNS connectivity. Additionally, many studies have demonstrated that genetic and environmental risk factors can directly cause cognitive impairment and demyelination in AD as well as in normal aging [16,17,18]. These studies suggest that the degradation of myelin may promote the accumulation of β-amyloid fibers, cause further myelin breakdown, and finally lead to neurodegeneration. Myelin protection is an important objective for the treatment of cognitive decline and dementia. Despite the high activity of SMase in the SAMP10 brain, the association between demyelination and neuronal function in these mice remains unclear.

Another relevant chemical species worth mentioning is plasmalogens (Pls), a subclass of cell membrane glycerolphospholipids abundant in many tissues, especially the heart, brain, and immune system [19,20]. Previous studies have demonstrated that Pls are located in specialized membranes, such as the sarcolemma [20], myelin [21], secreted membranes such as synaptic vesicles [21], and secretory granules. In mammals, Pls are mainly classified into two categories according to their head group: choline Pls and ethanolamine Pls (PlsEtn). The level of PlsEtn in brain tissue depends on the degree of myelination [21], and 70% of ethanolamine glycerophospholipids in the myelin sheath are composed of PlsEtn. The level of PlsEtn declines with normal aging, which is related to a decrease in the degree of myelination [22] and several neurodegenerative diseases, including Parkinson’s disease [23], Niemann-Pick type C disease [24], and dementia, such as AD [21]. Therefore, we hypothesized that an increase in Pl levels might lead to cognitive improvement in SAMP10 mice.

Furthermore, given the fact that fish oil (FO) and/or FP provide protective effects against age-related memory impairment and cognitive dysfunction, we investigated whether FO or FP feeding may prevent short-term memory loss, demyelination, and decreased levels of Pls in SAMP10 mice.

## 2. Materials and Methods

### 2.1. Materials and Reagents

FP was prepared as previously described [9]. Briefly, FP was extracted from Alaska pollock (*Theragra chalcogramma*) fillets. After freeze-drying the muscles (FDU-1200, EYELA, Tokyo, Japan) and washing with *n*-hexane/ethanol (1:1), the residues were air-dried, ground using a Waring blender (GM200; Retsch Technology GmbH, Haan, Germany), and then stored at −35 °C. The yield of FP (lipid content being less than 0.1%) from the fillets was 11.8% (*w*/*w*). FO was purchased from Nippon Suisan Kaisha, Ltd. (Tokyo, Japan). The AIN-93 vitamin mix, AIN-93G mineral mix, dextrinized cornstarch, cornstarch, cellulose, sucrose, and casein were purchased from Oriental Yeast Co., Ltd. (Tokyo, Japan). L-Cystine, choline bitartrate, and soybean oil were purchased from FUJIFILM Wako Pure Chemical Co. (Osaka, Japan). All other chemicals were obtained from common commercial sources and were reagent grade.

### 2.2. Nutritional Composition

The water, ash, crude protein, and crude fat contents of casein and FP were determined in accordance with the Official Methods of Analysis of AOAC International (Table 1) [25]. Amino acid compositions were determined by the Japan Food Research Laboratories (Tokyo, Japan; Table 1). Fatty acid compositions were determined using a gas chromatograph (GC) with flame ionization detection (GC-FID) (GC-2014; Shimadzu Co., Kyoto, Japan) equipped with an Omegawax^®^ capillary column (Merck KGaA, Darmstadt, Germany), as described previously (Table 2) [26].

### 2.3. Animals and Diets

One-month-old male SAMP10/TaSlc mice were purchased from Japan SLC, Inc. (Hamamatsu, Japan). Thirty mice were randomly divided into three diet-based groups: control, FP, and FO (10 mice per group). Animals were maintained on a 12-h light/dark cycle (lights on at 8:00 a.m.) in a temperature- and humidity-controlled specific-pathogen-free animal facility at Kansai University. The mice had *ad libitum* access to food and water. All animal experiments were approved by the Animal Ethics Committee of Kansai University. All experiments were conducted in accordance with the approved guidelines and regulations (approval number: 1917). After acclimating for five days by eating a diet prepared in accordance with the American Institute of Nutrition (AIN-93G) recommendations [27], the mice were divided into three groups and fed the experimental diets. As shown in Table 3, these experimental diets contained the essential composition based on the AIN-93G formula. Body weight (BW) and food intake were measured once per week. After five months, the mice were subjected to behavioral tests. The animals were euthanized under anesthesia to collect the blood, liver, spleen, kidney, cecum, and brain one week after the conclusion of all behavioral tests, to minimize their influence on serum biochemical parameters in accordance with previous studies [28,29].

### 2.4. Behavioral Tests

All behavioral tests were performed from 9:00 a.m. to 6:00 p.m. To reduce the influence of prior tests, behavioral tests were ordered from a low to high degree of stress. Additionally, the experiments were conducted in a sound-attenuated and air-regulated excremental room, and mice were habituated in this room for 1 h before testing.

#### 2.4.1. Y-maze Test

The spontaneous alternation behavior of mice in the Y-maze, an index of short-term memory, was assessed in accordance with the methods outlined in a previous report [28]. The Y-maze apparatus consisted of black-frosted Plexiglas, with each arm measuring 40 × 10 × 12 cm (L × W × H), tapering to 3 cm wide at the bottom. The arms converged to a triangular center of 4 cm per side. Each mouse was placed at the end of one arm and allowed to move freely throughout the maze during an 8-min session. The series of arm entries were visually observed and recorded. Spontaneous alternation behavior was defined as consecutive entry into all three arms (i.e., arm A, B, and C) in triplet sets (i.e., ABC, ACB, BAC, BCA, CAB, and CBA). Alternation behavior was calculated as the ratio of actual alternations to possible alternations (defined as (total number of arm entries – 2) × 100) and was presented as a percentage, as described previously [28].

#### 2.4.2. Novel-Object Recognition Test (NORT)

NORT was performed according to the method outlined in earlier reports [28,29]. The test consisted of three sessions: habituation, training, and retention. The habituation session consisted of a 10-min exploration of an acrylic cage with a gray-frosted Plexiglas floor (30 cm × 30 cm × 35 cm) without any objects for 2 days. During the training session, two objects were placed in the back corner of the box. The objects included wooden quadrangular prism, metal cylinder, and a golf ball, which were different in shape, color, and material, but similar in size. Each mouse was individually placed midway toward the front of the box, and the total time spent exploring the object was recorded for 10 min. A mouse was considered to be exploring the object when its head was facing the object or when it was touching or sniffing it. During the retention session, the mouse was placed back into the same cage 24 h after the training session; however, one familiar object from the training session was replaced with a novel object. The mice were then allowed to explore freely for 5 min, and the time spent exploring each object was recorded. Throughout the experiments, the objects were counterbalanced in terms of their physical complexity and emotional neutrality. The discrimination index, calculated as the ratio of the time spent exploring the novel object (retention session) to the total time spent exploring both objects, was used to measure cognitive function.

### 2.5. Sample Collection

The mice were deeply anesthetized with 2% isoflurane (FUJIFILM Wako Pure Chemical Co.) and blood was collected from the abdominal vena cava. The animals were then transcardially perfused with 0.9% saline. The entire brain was quickly removed and the hippocampus (Hipp) and cortex were isolated from the right hemisphere in ice-cold saline, and then immediately frozen. For histological analysis, the left side of the brain was fixed in 4% paraformaldehyde in phosphate-buffered saline (PBS) for 14–16 h at 4 °C. The fixed tissues were cryoprotected in PBS containing 20% sucrose for 24 h and 30% sucrose for 48 h until equilibrating, embedded in PolyFreeze compound (Cat.#25116-4; Polysciences Inc., Warrington, PA, USA), and then cut into 10-µm sections using a cryostat (Cat.#Leica CM1850; Leica Biosystems, Nussloch, Germany) for immunohistochemical analysis. All samples and sections were stored at −80 °C until analysis.

### 2.6. Immunohistochemistry

Immunofluorescence staining was performed. The coronal sections between 1.45 and 2.25 mm from bregma (Allen Brain Atlases) were incubated in 10 mM citrate buffer (pH 6.0) in a water bath at 80 °C for 30 min. After briefly washing with PBS, the sections were incubated with PBS containing 0.5% Triton X-100 for 15 min at 22–25 °C and then with blocking buffer (3% bovine serum albumin in Tris-buffered saline containing 0.05% Tween20; TBST) for 30 min at 22–25 °C. The sections were incubated at 4 °C for 14–16 h with primary antibodies against myelin basic protein (MBP) (1:3000; Cat#836504; BioLegend, Inc., San Diego, CA, USA) and a neurofilament heavy chain (NfH) (1:1000; Cat#AB5539; Merck KGaA, Darmstadt, Germany). The sections were washed three times with TBST for 10 min and then incubated with the appropriate secondary antibodies conjugated with Alexa 488 or Cy^TM3^ (Cat#103-545-155, Cat#115-165-003, Jackson ImmunoResearch Inc., West Grove, PA, USA; 1:500) for 2 h at 22–25 °C. Nuclei were stained with 4′,6-diamidino-2-phenylindole (DAPI, Cat#340-07971, FUJIFILM Wako Pure Chemical Co.). Sections were visualized under a Zeiss LSM700 confocal laser microscope (Carl Zeiss Microscopy, LLC, White Plains, NY, USA). Images were acquired using the ZEN black edition software version 16.0.0.0 (Carl Zeiss Microscopy, LLC). The immunohistochemical controls were performed as described above, except for the omission of primary antibodies. No positive immunostaining signals were observed in any of the control sections. The areas of positive cells for immunoreactivities were analyzed using the Fiji ImageJ software version 1.53c [30,31]. The average of two slices from at least 6 mice was used to calculate the percentage of the total analyzed area in the CA1 Hipp region. This was then used for statistical analysis.

### 2.7. Measurement of Serum Biochemical Parameters and the Levels of Plasmalogens (Pls) in Tissues

Serum was obtained by centrifugation (2000× *g* for 15 min at 4 °C). Biochemical parameters were analyzed at the Japan Medical Laboratory (Kaiduka, Japan). The level of Pls in brain tissues was determined according to previous studies [32]. Briefly, total lipids were extracted from the tissue through homogenization with chloroform-methanol (2:1, *v*/*v*) in accordance with the procedure described by Folch et al. [33]. The lipid-containing solvent was evaporated under a nitrogen atmosphere at room temperature (22–25 °C). One of the tubes served as a sample blank, and the other two were used for measuring Pl iodine uptake. After evaporation of the solvent, 0.9 mL of methanol and 3.2 mL of 0.094 M sodium citrate (pH 5.5) were added. Further, 0.9 mL of 3 M KI was added to the sample blank tube, and 0.4 mL of 3 M KI and 0.5 mL of 0.5 mM I_2_ in 3 M KI were added to the other two tubes separately, then incubated at room temperature for 40 min. Then, 5 mL of *n*-butyl acetate was added to each tube, which was then centrifuged at 2000× *g* for 10 min. Aliquots of *n*-butyl acetate were measured at an optical density of 363 nm against a reagent blank.

### 2.8. Statistical Analysis

All results are expressed as the mean ± standard error of the mean (SEM). Intergroup comparisons were conducted using one-way factorial analysis of variance (ANOVA), followed by the Tukey–Kramer multiple comparison test. GraphPad Prism version 8.4.3 (GraphPad Software, San Diego, CA, USA) was used for statistical analysis and generation of graphs. Statistical significance was set at *p* < 0.05.

## 3. Results

### 3.1. Effect of Diets Containing FO or FP on Growth Parameters, Relative Organ Weights, and Serum Biochemical Parameters

The 30 SAMP10 mice were randomized into three groups. After one month, three mice fed with FO died suddenly; however, there was no significant difference in survival rate between groups (data not shown). The body weight (BW) of each group increased significantly throughout the experimental period, from 19.1 ± 0.84 g to 34.4 ± 1.36 g for the control diet, 19.1 ± 0.68 g to 37.0 ± 1.51 g for the FO diet, and 18.9 ± 0.95 g to 35.5 ± 1.35 g for the FP diet (*p* < 0.0001 vs. initial BW for all groups, Table 4). There was no significant difference between the three groups with regard to BW, BW gain, and amount of daily food intake. There was also no significant difference in the relative organ weights (liver, spleen, kidney, and cecum) between the three groups. *Regarding serum biochemical parameters*, the concentrations of total protein and albumin were not significantly different among the three groups. No differences were found in the levels of aspartate aminotransferase (AST) and alanine aminotransferase (ALT) for liver function, creatine phosphokinase (CPK) for skeletal muscle damage, and lactate dehydrogenase (LDH) for cell damage. Serum total cholesterol and high-density lipoprotein cholesterol (HDL-C) levels were significantly lower in the FO group than in the control and FP groups (*p* < 0.0001, Figure 1). The FO diet also significantly decreased the levels of serum triglycerides (TG) compared to those in the FP group (*p* = 0.038). The levels of blood urea nitrogen (BUN) were significantly higher (*p* < 0.01); however, the amount of creatinine tended to be lower in the FO group than in the other two groups. The ratio of BUN/creatinine was also significantly higher than that in the other two groups (control, 194.1 ± 15.1; FO, 370.5 ± 43.2; FP, 147.7 ± 12.4; *p* < 0.0001). While the BUN levels did not change in the FP group compared to the control group, the serum creatinine level was significantly higher in the FP group than in the other groups (*p* < 0.001). The BUN/creatinine ratio did not differ significantly from that of the control diet.

### 3.2. Effect of Diets Containing FO or FP on Short-Term Memory

The Y-maze test was performed to investigate the effects of diets containing FO or FP on the prevention of short-term memory loss in SAMP10 mice (Figure 2a). In the Y-maze test, spontaneous activity, as indicated by the total number of arm entries, was not significantly different between the groups. However, working memory, as indicated by the percentage of spontaneous alternation behaviors, was significantly higher in the FP group than in the other two groups (*p* = 0.0036 vs. the control diet, *p* = 0.0019 vs. the FO diet, Figure 2a).

Episodic memory was evaluated using NORT. In this test, the mice that were fed the control diet displayed a discrimination index lower than 50%, indicating that animals did not remember having previously known the old object (Figure 2b). While FO and FP-fed mice showed an index higher than 50%, there was no significant difference in the discrimination index between the experimental groups.

### 3.3. Effect of Diets Containing FO or FP on the Morphological Changes in the CA1 Hipp Region

The NfH immune-positive area in the CA1 Hipp region of 6-month-old SAMP10 mice fed the FP diet was significantly higher than that in the control group (*p =* 0.004 vs. control group, Figure 3a,b) and higher, but not statistically significant, in the FO group compared to the control group (*p =* 0.202 vs. control group, Figure 3a,b). The MBP immune-positive area in the same region with the FP diet was greater than that in the control or FO group, but the difference was not statistically significant. The percentage of MBP/NfH double-positive area was significantly higher in the FP group, compared to the control group and FO group *(p =* 0.016 vs. control group, *p =* 0.075 vs. FO group, Figure 3a,b). This suggests that FP feeding protected the myelin sheath in the CA1 Hipp region in 6-month-old SAMP10 mice, better than the control and FO diet.

### 3.4. Effect of Diets Containing FO or FP on the Levels of Pls in an SAM Brain at 6 Months of Age

Pls are alkenyl-acyl glycerophospholipids. Previous studies have demonstrated that Pls can act as antioxidants, and their levels decreased in the brains of post-mortem AD patients. Therefore, we determined the Pls levels in the brains of SAMP10 mice. The findings showed a significant increase in Pls levels in the Hipp of mice fed the FO diet compared to the control diet; however, there was no significant difference in FP-fed mice (*p* = 0.049 vs. control group, *p* = 0.0076 vs. FP group, Figure 4). Furthermore, there was no significant difference in the cortex between the groups.

## 4. Discussion

Recent studies have suggested that regular consumption of fish may be beneficial for human health and brain function. In this study, we investigated the protective roles of FO and FP against memory impairment and cognitive dysfunction in SAM brains. We also examined whether FO or FP feeding improved demyelination and lowered the levels of Pls in SAMP10 mice. Our study demonstrated that FP feeding significantly prevented short-term working memory loss and demyelination in SAMP10 mice. While FO feeding significantly decreased the lipid levels in the serum and increased those of Pls in the Hipp, it was not effective in protecting against memory loss in SAMP10 mice.

The weekly food intake, growth parameters, and serum enzyme levels of animals in all groups were not affected by dietary treatments. The activities of enzymes (AST, ALT, CPK, and LDH) in the serum are generally used to diagnose impairment and damage in the heart, liver, skeletal muscle, and kidney. In this study, the mean values of all tested serum enzymes were not significantly different regardless of the diet. Therefore, these findings clearly demonstrated that exposing animals to diets containing FP or FO for a long period will cause no adverse effects on the functioning of their vital organs, including the liver and kidney. In addition, our findings are consistent with those of earlier studies which found that dietary FP or FO decreased the serum levels of total cholesterol, TG, and low-density lipoprotein cholesterol in rats [34,35]. One study also demonstrated that the FO diet significantly reduced total cholesterol and HDL-C in mouse plasma [36]. However, there are limited studies that report a link between FP diets and blood cholesterol levels. Likewise, our study could not find evidence of the beneficial effects of FP on serum lipid levels; however, FO was found to decrease lipid concentrations.

The Y-maze evaluates spatial working memory, an index of short-term memory; therefore, this test is considered to be an assessment of Hipp-dependent memory [37]. In contrast, with a longer intersession interval, NORT evaluates long-term memory usually 24 h between training and test sessions [38]. A previous study showed that when rodents are exposed to familiar and novel objects, they spend more time approaching the novel object than the familiar one [39]. Recognition memory is related to the olfactory cortex, which is considered one of the major neuronal information pathways for episodic memory [40]. It has been observed that the learning and memory abilities of SAMP10 mice decline earlier than those of normally aging mice [41]. Hashimoto et al. demonstrated that 5-month-old SAMP10 mice showed no significant changes in spontaneous alternations in the Y-maze test or exploratory preference in the NORT compared to control mice [41]. However, we observed a decrease in short- and long-term memory in 5-month-old SAMP10 mice. This may be because the experimental diets were composed of the minimum amounts of essential and semi-purified nutrients based on AIN-93G, compared to diets made with non-purified components (CE-2 diet; Clea Japan Inc., Yokohama, Japan). Previous research has indicated that host behavior and gut microbiotas are significantly affected by age and purity of dietary components, that is, either non-purified or semi-purified materials [42]. Therefore, our experimental diet provided stricter conditions for SAMP10 mice, and the results suggest that FP intake provides beneficial effects in maintaining brain function.

A decrease in the weight and cortical thickness of the brain in SAMP10 mice has been observed at 5 months of age, with further significant reductions at 8 months [43,44]. Age-related brain atrophy in SAMP10 mice has been reported mainly in the olfactory bulbs and frontal cortex [44], which also commonly occurs in the elderly human brain, especially in certain neurodegenerative diseases with dementia. Additionally, it has been demonstrated that the levels of β-amyloid precursor protein increase in an age-dependent manner in the Hipp of SAMP10 mice [45]. The pyramidal neurons in the Hipp are known as “place cells” for spatial memory [46], which reactivate specific memory representations of the olfactory cortex and amygdala during memory retrieval [47]. SMase, an enzyme that generates sphingolipid ceramides, is an important effector in the brain. It has been reported that the levels of SMase activity are remarkably increased in the aged rat brain [48], and their inhibition may improve the quality of myelin and stabilize its structure [13]. SMase activity in the cerebral cortex of SAMP10 mice at 10 and 17 months of age was found to be significantly higher than that in mice at 2 months of age [14]. Furthermore, alterations in myelination were discovered to be important pathophysiological correlates of AD and MCI [16,17]. We demonstrated that the level of demyelination in the Hipp was significantly decreased by FP feeding, compared to the control and FO diets. Therefore, we assume that one of the mechanisms for preventing short-term memory loss may be related to the inhibition of SMase activity and protection against myelin breakdown. However, previous studies have shown that the levels of Pls were decreased in several neurological disorders, including AD [49], Parkinson’s disease [50], multiple sclerosis [51], and cerebrovascular ischemia [52]. We measured the Pl levels in the Hipp and cortex; while the FO diet did increase the Pl levels in the Hipp, no evidence of significant changes was found with the FP diet. Therefore, the increased levels of Pls in the Hipp may not be linked to the prevention of short-term memory decline and demyelination. However, it is worth noting that prior studies on the effects of food protein-derived components have shown that soy peptides also played a role in preventing age-dependent cognitive impairment. This was achieved via an increased expression of neurotrophic factors, such as brain-derived neurotrophic factor and neurotrophin-3, in SAM brains [53]. Other research studies have also demonstrated that carnosine, an endogenous antioxidant peptide extracted from chicken, prevents cognitive decline caused by an increase in cerebrovascular alterations and microglial activation in the Hipp of AD transgenic mice fed a high-fat diet [54]. Therefore, we hypothesize that the physiological activities of FP which protected SAMP10 mice against short-term memory impairment may depend on certain unique antioxidant peptides present in FP. Our report may be limited because the underlying mechanisms of the FP diet in the protection of the mouse brain and functional peptides and/or components remain unclear. However, our study demonstrated that FP preserved the axonal components of mouse neuronal cells. Further research is needed to prove FP indeed improves cognitive function based on the tendency found in this study, and more studies on brain-gut interaction will help shed light on the molecular and biological mechanism of FP to protect brain function.

## 5. Conclusions

Our findings suggest that FP prevents age-related short-term memory loss in SAMP10 mice. The results also indicate that FP intake provides physiological benefits in maintaining axon morphology in the Hipp of SAMP10 mice. The mice showed morphological changes similar to those observed in the brains of elderly humans with MCI [55]. Consequently, dietary fish intake, especially FP, could significantly reduce the risk of dementia associated with aging. A more in-depth investigation, however, is required to explore the nature of the underlying mechanisms concerning FP in the prevention of age-related mental impairment.

## Figures and Tables

**Figure 1 nutrients-14-04618-f001:**
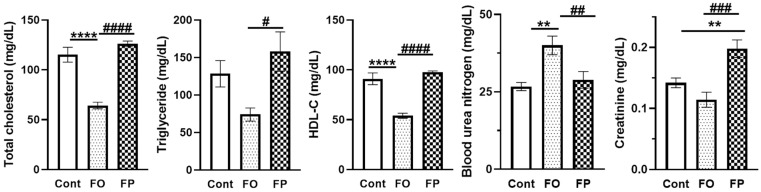
Effects of a diet containing fish oil (FO) or fish protein (FP) on serum biochemical parameters in a senescence-accelerated mouse model at 6 months of age. All data are presented as the mean ± standard error of the mean (*n* = 10 mice for control and FP, *n* = 7 for FO). ** *p* < 0.01, **** *p* < 0.0001 vs. control diet group (Cont = control), ^#^ *p* <0.05, ^##^ *p* < 0.01, ^###^ *p* < 0.001, ^####^ *p* < 0.0001 vs. FO diet group. HDL-C = high-density lipoprotein cholesterol.

**Figure 2 nutrients-14-04618-f002:**
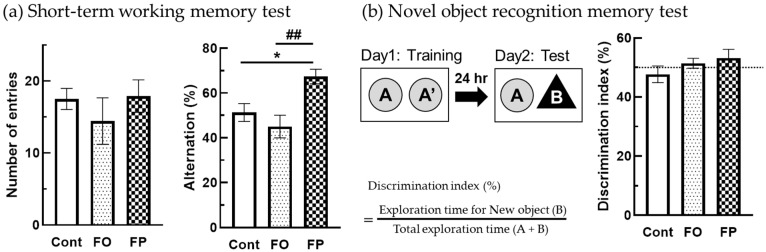
Fish protein (FP) feeding prevented loss of short-term memory in a senescence-accelerated mouse model at 6 months of age, better than the control diet or fish oil (FO) feeding. (**a**) Spontaneous activity and working memory in the Y-maze test. Total arm entries and alternation behavior were measured during an 8-min session. (**b**) Object recognition memory was measured in a novel object-based recognition test. A memory retention session was performed 24 h after the training session. The discrimination index was calculated as described. The dotted line represents 50% of discrimination index levels. All data are presented as the mean ± standard error of the mean (*n* = 10 mice for control and FP, *n* = 7 for FO). * *p* < 0.05 vs. control diet group (Cont = control), ^##^ *p* < 0.01 vs. FO group.

**Figure 3 nutrients-14-04618-f003:**
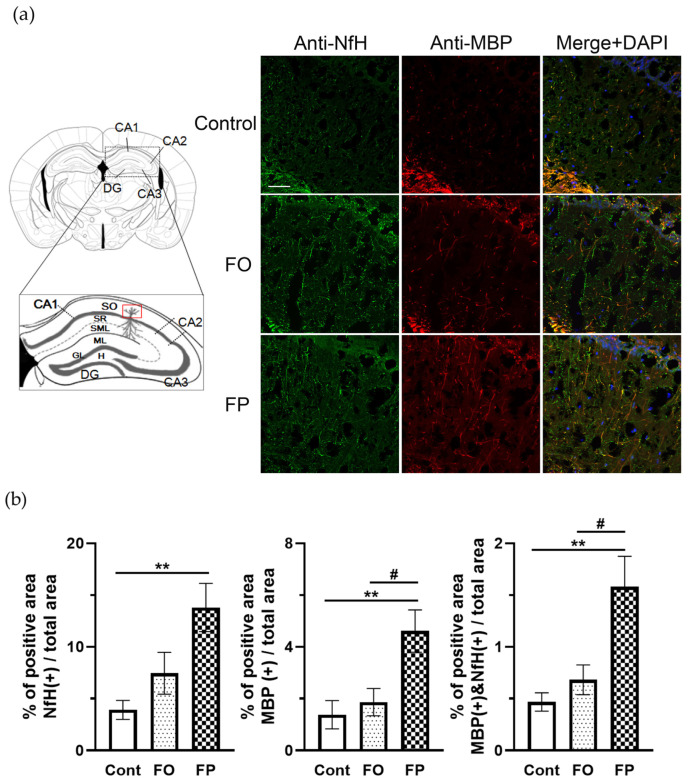
Fish protein (FP) feeding protected the myelin sheath in 6-month-old senescence-accelerated mouse CA1 region of hippocampus (Hipp). (**a**) Schematic illustration showing the Hipp section that was included in region for visualizing (red box). Images of double staining with neurofilament heavy chain (NfH; green) and myelin basic protein (MBP; red) in the CA1 region of Hipp. The scale bar denotes 50 µm. DAPI = 4′,6-diamidino-2-phenylindole. (**b**) NfH or MBP positive area, and MBP/NfH double-positive area per total analyzed area were quantified in the CA1 Hipp region. Data are presented as the mean ± standard error of the mean (the average of two sections from at least 6 mice per group). ** *p* < 0.01 vs. control diet group, ^#^ *p* < 0.05 vs. FO group. DG = dentate gyrus; SO = stratum oriens; SR = stratum radiatum; SLM = stratum lacunosum; ML = molecular layer of dentate gyrus; GL = granule cell layer; H = hilus of dentate gyrus; Cont = control.

**Figure 4 nutrients-14-04618-f004:**
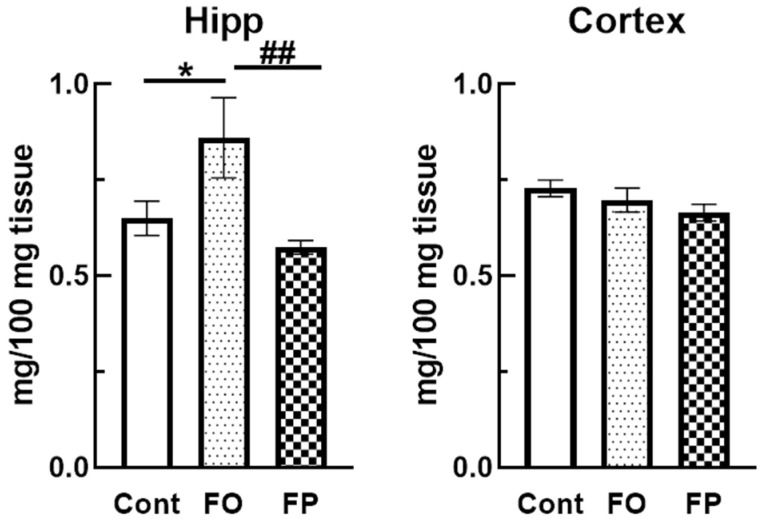
Fish oil (FO) feeding significantly increased the level of plasmalogens in a senescence-accelerated mouse hippocampus (Hipp) at 6 months of age, compared to the control or fish protein (FP) diets. Data are presented as the mean ± standard error of the mean (*n* = 10 mice for control and FP, *n* = 7 for FO). * *p* < 0.05, vs. control diet group (Cont = control), ^##^ *p* < 0.01 vs. FO group.

**Table 1 nutrients-14-04618-t001:** Nutritional and amino acid composition of casein and fish protein (FP).

	Experimental Proteins	
	Casein	FP
Water (g/100 g)	2.5	2.5
Ash (g/100 g)	1.6	4.6
Crude protein (g/100 g)	93.9	92.1
Amino acid composition (wt%)		
Alanine	3.07	6.40
Arginine	3.56	6.65
Aspartic acid ^1^	6.81	10.64
Cysteine	0.38	1.11
Glutamic acid ^2^	21.07	16.19
Glycine	1.81	4.63
Histidine	2.67	2.04
Isoleucine	5.23	4.99
Leucine	9.03	8.65
Lysine	8.04	10.14
Methionine	2.55	3.31
Phenylalanine	4.88	4.16
Proline	10.6	3.36
Serine	4.92	4.27
Threonine	3.85	4.42
Tyrosine	5.30	3.72
Valine	6.23	5.32
Crude fat (g/100 g)	1.3	0.5
EPA + DHA (g/100 g)	N.D.	0.1

Aspartic acid ^1^ = aspartic acid and asparagine. Glutamic acid ^2^ = glutamic acid and glutamine. EPA = eicosapentaenoic acid. DHA = docosahexaenoic acid. N.D. = not detected.

**Table 2 nutrients-14-04618-t002:** Fatty acid composition of soybean oil and fish oil (FO).

	Experimental Oils	
	Soybean Oil	FO
Fatty acid composition (wt%)		
C14: 0	N.D.	8.8
C16: 0	11.3	18.2
C16: 1	N.D.	13.1
C18: 0	3.5	3.1
C18: 1*n*-9	21.8	6.9
C18: 1*n*-7	N.D.	3.5
C18: 2*n*-6	53.6	1.6
C18: 3*n*-3	5.5	1.4
C20: 4*n*-6 (AA)	N.D.	1.1
C20: 5*n*-3 (EPA)	N.D.	16.4
C22: 6*n*-3 (DHA)	N.D.	11.7
Others	4.3	14.2

N.D. = not detected. AA = arachidonic acid. EPA = eicosapentaenoic acid. DHA = docosahexaenoic acid.

**Table 3 nutrients-14-04618-t003:** Composition of experimental diets (g/kg).

Components	Experimental Groups		
	Control	FO	FP
Casein	200	200	—
Fish protein	—	—	203.9
Soybean oil	67.4	20	68.98
Fish oil	—	50	—
*tert*-Butylhydroquinone	0.014	0.014	0.014
Corn starch	400.1	397.5	394.6
Dextrinized corn starch	132	132	132
Sucrose	100	100	100
Cellulose	50	50	50
L-Cystine	3	3	3
Choline bitartrate	2.5	2.5	2.5
AIN-93 vitamin mixture	10	10	10
AIN-93G mineral mixture	35	35	35

Diets were prepared based on the composition of AIN-93G. The reason for different amounts of protein, soy oil, and corn starch in each experimental group is to align the amount of protein, fat, and starch in the diets. Because of the crude contents, the amount of protein, corn starch, and soybean oil are slightly different in each diet. FO = fish oil group. FP = fish protein group.

**Table 4 nutrients-14-04618-t004:** Growth parameters, relative organ weights, and serum biochemical parameters of 6-month-old mice fed the experimental diets for five months.

	Experimental Groups		
	Control	FO	FP
**Growth parameters**			
Initial BW (g)	19.1 ± 0.84	19.1 ± 0.68	18.9 ± 0.95
Final BW (g)	34.4 ± 1.36	37.0 ± 1.51	35.5 ± 1.35
BW gain (g/day)	0.10 ± 0.01	0.12 ± 0.01	0.11 ± 0.01
Food intake (g/day)	4.87 ± 0.17	4.97 ± 0.23	5.24 ± 0.16
**Organ weights (g/100 g BW)**			
Liver	4.12 ± 0.17	4.63 ± 0.09	4.38 ± 0.16
Spleen	0.20 ± 0.01	0.20 ± 0.01	0.19 ± 0.02
Kidney	1.84 ± 0.08	1.93 ± 0.14	1.98 ± 0.11
Cecum	0.71 ± 0.07	0.80 ± 0.06	0.68 ± 0.06
**Serum biochemical parameters**			
Total proteins (g/dL)	4.86 ± 0.08	4.83 ± 0.06	4.69 ± 0.06
Albumin (g/dL)	2.86 ± 0.06	2.84 ± 0.04	2.74 ± 0.03
LDH (U/L)	191.7 ± 8.94	186.7 ± 19.0	206.8 ± 15.1
AST (U/L)	46.3 ± 1.72	50.7 ± 2.08	48.6 ± 1.69
ALT (U/L)	21.2 ± 2.23	30.9 ± 3.24	27.7 ± 2.79
CPK (U/L)	24.3 ± 3.04	27.4 ± 2.61	26.2 ± 3.24

FO = fish oil. FP = fish protein. BW = body weight. LDH = lactate dehydrogenase. AST = aspartate aminotransferase. ALT = alanine aminotransferase. CPK = creatine phosphokinase. All data are presented as the mean ± standard error of the mean (n = 10 mice for control and FP, n = 7 for FO).

## Data Availability

The data presented in this study are available on request from the corresponding author, upon reasonable request.

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
