# Peer review of "Protective Effects of Fish (Alaska Pollock) Protein Intake against Short-Term Memory Decline in Senescence-Accelerated Mice"

_nutrients, 2022, doi:10.3390/nu14214618_

Round 1
Reviewer 1 Report
The study focuses on the potential beneficial effects of consuming protein and fish oil.
The study was actually very well designed, and it would seem that it could provide important information. However, even though the introduction is fine with the possibility of improvement, the methods and therefore the results leave me quite uneasy. There are methodological flaws from the number of animals used for behavioral studies, to the techniques of tissue preservation and cutting. In turn, in the conclusions it seems to me that they make assumptions beyond the results obtained.
I have made several comments in the pdf file of the article that should be addressed in order to be published.

Author Response
RESPONSES TO REVIEWER COMMENTS
Reviewer #1:
We thank the Reviewer for their constructive comments. We have added our replies and explanations, immediately after each comment. We have yellow highlighted the corrections in the text for clarity.
Comments:
(1) L32-34, I think the idea and abstract should talk about neuroprotective effects found in food rather than toxic effects to give sequence to the information that is mentioned later.
Response:
As suggested, we focused on the neuroprotective effects of food (p. 1, L32-35).
(2) L119, How many animals were treated?
Response:
We added the number of animals treated for each experimental group in the revised manuscript (p. 4, L221-222). Thirty animals were randomly divided into three groups, with 10 mice per group.
(3) L131, The animals were
Response:
We replaced “They” with “The animals” as per the reviewer’s suggestion (p. 4, L232).
(4) L132, specify which tissues were obtained
Response:
Based on this suggestion, we specified which tissues were obtained (p. 4, L233). We measured the weight of the liver, spleen, kidneys, cecum, and brain.
(5) L132, Justify why the animals were not sacrificed right after the tests
Response:
As mentioned in L236-237, the reason of delayed euthanization was to minimize the influence of the behavioral tests on serum biochemical parameters. This was done in accordance with previous studies. We added references to these studies to justify this choice (p. 4, L236-237, references 28, 29).
(6) L177-178, please clarify if the tissues prior to freezing were dehydrated
Response:
For clarity, we added more details regarding the dehydration procedure (p. 6, L317-319).
(7) L220-223, It is not mentioned what type of normality tests were performed nor what test to determine the difference between the means (ANOVA test) or a comparison of the mean ranges (Kruskal Wallis). When Tukey's test is mentioned, I assume that it is a complement to ANOVA, however it is important to clarify.
Response:
We performed a one-way factorial ANOVA first and confirmed equal variances across samples and groups by ANOVA (Bartlett's test and Brown–Forsythe test). We corrected the related methods (p. 7, L417-418).
(8) L234, it does not mentioned if its serum or plasma albumin
Response:
As we mentioned at the beginning of the sentence (p. 7, L432-433), these are the results “regarding serum biochemical parameters”.
(9) L252, How many animals per group? Explain why the difference between the number of each group
Response:
Based on this comment, we added the number of animals for each experimental group (p. 4, L223-224). As shown in the Results (p. 7, L424-426), three mice fed with FO died suddenly during the experimental period. Therefore, the final number was different between the groups (p. 8, L464).
(10) L267, episodic memory
Response:
We corrected “episode memory” with “episodic memory,” as suggested (p. 8, L479).
(11) L268-269, In fact, what the test expresses is that the animal does not remember having previously known the old object, and therefore it assumes them both as if they were new objects and spends the same time recognizing one or the other.
Response:
In accordance with the reviewer’s observation, we modified “they did not recognize the novel object” with “the animals did not remember having previously known the old object” (p. 8, L480-481).
(12) L282, why the n of one of the groups reduced to 6.? Methodologically, there is an important error because memory and learning tests require at least n of 8 animals per group due to the great statistical variability in these tests.
Response:
As we mentioned for comment (9), three animals of the FO group died. We also excluded those animals that entered less than 10 arms in total in the Y-maze test. In the revised manuscript, we included all the animals tested. Nevertheless, the Y-maze test results remained statistically significant; thus, we assume that the conclusions drawn from the test results do not change (Figure 2).
(13) L297, It would seem that the tissue used have processing problems. Significant holes are observed in the tissues which may be due to poor dehydration or the knife with which the cuts were made.
Response:
All samples were similar between groups. In the SAMP10 mouse brain, however, neuronal death was observed at an early age, and many holes could be found in the brain sections, as also reported in previous studies (e.g., Nomura Y et al., J Pharmacol Sci. 92(1):28-34. doi: 10.1254/jphs.92.28. PMID: 12832852.). In addition, all tissues were prepared at the same time with same methods. Therefore, we can assume that the holes present were not due to poor dehydration.
(14) L300, it appears to be more cortex than hippocampus. it appears to be more cortex than hippocampus
Response:
The tissue presented is the upper portion of the pyramidal cell layer, namely the stratum oriens (SO) of the CA1 hippocampal region, a key region for memory which is rich in neuronal axons (Kim J et al., J Neurosci. 38(22):5140-5152. doi: 10.1523/JNEUROSCI.0155-18.2018. PMID29728449; Kwakowsky A et al., J Neurochem.
155(1):62-80. doi: 10.1111/jnc.15099. PMID32491248). We added a schematic illustration of the hippocampal section that we analyzed to Figure 3.
(15) L334, Heart has not been mentioned before
Response:
As mentioned in our response to comment (4), we did not measure the weight of the heart. However, seral LDH is generally used to diagnose heart damage. We modified the sentence for clarity (p. 11, L719).
(16) L411-412, Axonal morphology seems to me too risky to mention when only some proteins are seen. To be able to talk about morphology, I think it should be seen with other microscopy techniques.
Response:
As we mentioned in our answer to comment (13), all tissues were prepared at the same time with same methods. However, we found normal axonal morphology only in mice fed with FP. Therefore, we assume that morphology is not linked to the procedure. To address this point, we increased the number of animals and re-analyzed our data; however, the results lead to the same conclusions (Figure 3b).

Reviewer 2 Report
The authors present a novel and relevant manuscript focused on the protective effects of fish protein (FP) and fish oil (FO) intake against cognitive decline memory in senescence-accelerated mice (SAM).
Overall the study is well described, however, the authors propose a protective effect of FO that is not proved in this work. The authors investigated whether FO or FP feeding prevented short-term memory loss in SAMP10 mice, and the Y-maze test results do not show statistical significantly higher working memory in FO group vs control group. Therefore, the authors should mitigate the interpretation of the results and their conclusions.
Similarly, the episode memory results are not well interpreted. In my opinion there is no difference in discrimination index between the groups (control, FO and FP). FP and FO groups have no shown better performance. Again, the authors should mitigate the interpretation of the results and their conclusions. If the results not provide statistically significant benefits, a tendency towards better performance cannot prove protective effects of FO and FP intake against episode memory in these animals. The authors propose a protective effect of FO and FP in episode memory that is not proved in this work.
Regarding morphological changes in the CA1 region of Hipp in FO group, the same happens. Immune-positive areas in FO diet group are not significantly higher compared to control group. These FO feeding results do not prove a protective effect in the myelin sheath.
In Statistics to be fairly close to the statistical significant value (p<0.05) mean the same as to be far. In both cases the null hypothesis is true, and therefore, we cannot conclude that there are differences. A tendency towards better data or results close to statistical significance do not prove an effect.
In any case, the authors cannot conclude with the results of this study that FO feeding slightly inhibited demyelination (FO does not inhibit it), or that it was less effective than FP in protecting against short- term working memory loss and episode memory (FO does not protect against them) in SAMP10 mice. Moreover, authors cannot conclude that FP protect against cognitive decline in SAMP10 mice.
First of all, the authors should modify the information related with the above exposed in the entire manuscript.
In my opinion, the following changes should be considered:
Line 1: …“and fish oil”… should be deleted.
Line 19-20: “FP-fed mice demonstrated a tendency toward better performance than the control (p = 0.06) in the novel object recognition test” should be deleted. Discrimination index between the groups are similar (control, FO and FP).
Line 22-23: “Axon morphology was maintained in FP and FO groups, but not in control” Should be modified. FO neither maintain axon morphology.
Line 25: … “FP intake is more effective than FO in preventing” Should be modified. FO has no effect in preventing.
Line 269-272: “Whereas, FO- and FP-fed mice showed better performance, despite the fact that there was no statistical significance (p = 0.335 vs. FO, p = 0.059 vs. FP). While FO intake did not provide any significant benefits, the results from FP intake indicated a tendency towards better performance due to being fairly close to the significant value”. Should be changed by “There was no significant difference in discrimination index between the groups”.
Line 276: … “better than casein and soybean oil or fish oil (FO) feeding” Should be deleted.
Line 299: “Fish oil (FO) or”… should be deleted. FO has no a proved effect in myelin sheath in this study.
Line 329-330: …”slightly inhibited demyelination, it was less effective than FP in protecting against short-term working memory loss and cognitive dysfunction”. Should be deleted.
Line 364: … “results suggested that FP intake provided more beneficial effects than FO intake in maintaining brain function”. Should be changed by “results suggested that FP intake provided beneficial effects in maintaining brain function”.
Line 381: “We demonstrated that the levels of demyelination in the Hipp were significantly decreased by FP feeding and were slightly maintained by FO feeding, compared to the control diet”. Should be changed by “We demonstrated that the levels of demyelination in the Hipp were significantly decreased by FP feeding compared to the control diet”.
Line 401-408: This paragraph is confused. The conjecture that authors done is not clear. Again, they continue discussing FO effects but this work does not prove the FO effects mentioned.
Line 404: …“FP maintained axonal components of mouse neuronal cells to a better degree than FO” should be changed by “FP maintained axonal components of mouse neuronal cells”.
Line 405-406: … “mechanisms of each diet in the protection of mouse brain function remain unclear” should be changed by “mechanisms of FP diet in the protection of mouse brain function remain unclear”.
Line 407-408: …“ further research is warranted to investigate the exact physiological roles played by FP and FO in preventing cognitive decline in SAMP10 mice”. Should be modified. Future studies should be done to prove the tendency found to improve short-term memory of FO in this study.
Line 410-411: “Our findings suggest that FP and FO prevent the decline of cognitive function and loss of short-term memory related to aging in SAMP10 mice” should be changed by “Our findings suggest that FP prevents the loss of short-term memory related to aging in SAMP10 mice”.
Line 411-412: “The results also indicate that FP intake provides greater physiological benefits than FO in maintaining axon morphology in the Hipp of SAMP10 mice” should be changed by “The results also indicate that FP intake provides physiological benefits in maintaining axon morphology in the Hipp of SAMP10 mice”.
Line 417: … “underlying mechanisms concerning FO and FP in the prevention of age-related mental impairment” should be changed by “ …“underlying mechanisms concerning FP in the prevention of age-related mental impairment”.
Other modifications proposed:
Lines 48-49-69-181: Please, include the symbols alpha, beta and mu.
Line 134: The composition of diets is well documented. Table 3 includes the composition of the experimental diets. In my opinion the differences between the 3 diets in the table 3 should be described: Why fish protein is 203.9 g/kg in the FP group and casein is 200g/kg in control and FO groups. The differences in g/kg of soybean oil in each group (67.4, 20 and 68.98) or corn starch (400.1, 397.5 and 294.6 g/kg).
Lines 257-283-307-320: The description of the control diet does not seem adequate. The FO diet also contains casein and all three diets contain soyabean oil. In materials and methods, it has already been described, in my opinion it is not necessary to indicate anything.
Line 318: …“compared to casein and soybean oil”… should be changed by “control diet, for example”.

Author Response
RESPONSES TO REVIEWER COMMENTS
Reviewer #2:
We thank the Reviewer for their constructive comments. We provide our replies and explanations in blue, immediately after each comment. In accordance with the Reviewer’s suggestions, we modified our entire manuscript and deleted the parts inherent to the protective effects of FO.
Comments:
(1) Line 1: …“and fish oil”… should be deleted.
Response:
We modified the title as suggested, in accordance with our findings.
(2) Line 19-20: “FP-fed mice demonstrated a tendency toward better performance than the control (p = 0.06) in the novel object recognition test” should be deleted. Discrimination index between the groups are similar (control, FO and FP).
Response:
In accordance with this observation, we have deleted the sentence.
(3) Line 22-23: “Axon morphology was maintained in FP and FO groups, but not in control” Should be modified. FO neither maintain axon morphology.
Response:
In accordance with this comment, we have revised the sentence to:
“Axon morphology was maintained in the FP group, but not in the control and FO groups” (p. 1, L22-23).
(4) Line 25: … “FP intake is more effective than FO in preventing” Should be modified. FO has no effect in preventing.
Response:
Accordingly, we modified the sentence as follows:
“FP intake prevents age-related cognitive dysfunction” (p. 1, L25-26).
(5) Line 269-272: “Whereas, FO- and FP-fed mice showed better performance, despite the fact that there was no statistical significance (p = 0.335 vs. FO, p = 0.059 vs. FP). While FO intake did not provide any significant benefits, the results from FP intake indicated a tendency towards better performance due to being fairly close to the significant value”. Should be changed by “There was no significant difference in discrimination index between the groups”.
Response:
The sentence was modified as suggested by the reviewer (p. 9, L282-283).
(7) Line 276: … “better than casein and soybean oil or fish oil (FO) feeding” Should be deleted.
Response:
The sentence was revised with: “Fish protein (FP) feeding prevented loss of short-term memory in a senescence-accelerated mouse model at 6 months of age, better than the control diet or fish oil (FO) feeding.” (p. 9, L479-481).
(8) Line 299: “Fish oil (FO) or”… should be deleted. FO has no a proved effect in myelin sheath in this study.
Response:
As per the reviewer’s suggestion, “Fish oil (FO) or” was deleted (p. 10, L672).
(9) Line 329-330: …”slightly inhibited demyelination, it was less effective than FP in protecting against short-term working memory loss and cognitive dysfunction”. Should be deleted.
Response:
In accordance with this suggestion, the sentence was deleted and replaced with: “it was not effective in protecting against memory loss in SAMP10 mice” (p. 11, L713)
(10) Line 364: … “results suggested that FP intake provided more beneficial effects than FO intake in maintaining brain function”. Should be changed by “results suggested that FP intake provided beneficial effects in maintaining brain function”.
Response:
We modified the sentence accordingly (p. 12, L777-778).
(11) Line 381: “We demonstrated that the levels of demyelination in the Hipp were significantly decreased by FP feeding and were slightly maintained by FO feeding, compared to the control diet”. Should be changed by “We demonstrated that the levels of demyelination in the Hipp were significantly decreased by FP feeding compared to the control diet”.
Response:
We modified the sentence accordingly (p. 12, L794-796). In accordance with the suggestion of Reviewer #1, we increased the number of animals and re-analyzed our data. The results for the levels of demyelination in the Hipp were significantly decreased by FP feeding compared to the FO and control diets (Figure 3).
(12) Line 401-408: This paragraph is confused. The conjecture that authors done is not clear. Again, they continue discussing FO effects but this work does not prove the FO effects mentioned.
Response:
In accordance with this suggestion, we revised the paragraph (p. 12, L802-804, L811-812) and excluded one reference (original reference number was #53). Therefore, the reference number from 54 to 56 was changed accordingly.
(13) Line 404: …“FP maintained axonal components of mouse neuronal cells to a better degree than FO” should be changed by “FP maintained axonal components of mouse neuronal cells”.
Response:
The sentence was modified accordingly (p. 12, L412-413).
(14) Line 405-406: … “mechanisms of each diet in the protection of mouse brain function remain unclear” should be changed by “mechanisms of FP diet in the protection of mouse brain function remain unclear”.
Response:
The sentence was modified accordingly (p. 12, L814-815).
(15) Line 407-408: …“ further research is warranted to investigate the exact physiological roles played by FP and FO in preventing cognitive decline in SAMP10 mice”. Should be modified. Future studies should be done to prove the tendency found to improve short-term memory of FO in this study.
Response:
We thank the reviewer for this comment; however, the comment is a bit confusing, given that one of the reviewer #2 comments stated that the reviewer denied the tendency towards a better data. Moreover, we did not prove the tendency to improve short-term memory of FO in this study. Therefore, we feel that your comment is a misinterpretation of the results and the sentence has been modified as: “further research is needed to prove FP indeed improves cognitive function based on the tendency found in this study, and more studies for brain-gut interaction will help shed light on the molecular and biological mechanism of FP to protect brain function” (p. 12, L817-820).
(16) Line 410-411: “Our findings suggest that FP and FO prevent the decline of cognitive function and loss of short-term memory related to aging in SAMP10 mice” should be changed by “Our findings suggest that FP prevents the loss of short-term memory related to aging in SAMP10 mice”.
Response:
The sentence was modified accordingly (p. 13, L993-994).
(17) Line 411-412: “The results also indicate that FP intake provides greater physiological benefits than FO in maintaining axon morphology in the Hipp of SAMP10 mice” should be changed by “The results also indicate that FP intake provides physiological benefits in maintaining axon morphology in the Hipp of SAMP10 mice”.
Response:
The sentence was modified accordingly (p. 13, L994-995).
(18) Line 417: … “underlying mechanisms concerning FO and FP in the prevention of age-related mental impairment” should be changed by “ …“underlying mechanisms concerning FP in the prevention of age-related mental impairment”.
Response:
The sentence was modified accordingly, and FO deleted (p. 13, L999-1000)
Other modifications proposed:
(19) Lines 48-49-69-181: Please, include the symbols alpha, beta and mu.
Response:
We thank the reviewer for this observation, we carefully revised the symbols to make sure they are maintained during conversion to PDF.
(20) Line 134: The composition of diets is well documented. Table 3 includes the composition of the experimental diets. In my opinion the differences between the 3 diets in the table 3 should be described: Why fish protein is 203.9 g/kg in the FP group and casein is 200g/kg in control and FO groups. The differences in g/kg of soybean oil in each group (67.4, 20 and 68.98) or corn starch (400.1, 397.5 and 294.6 g/kg).
Response:
The reason for the different amounts of protein, soy oil, and corn starch sources in the control, FO, and FP diets is to align the amount of protein, oil, and starch in the diets. As described in the Methods and Table 1, the crude protein and fat contents of casein and FP differ slightly (93.9 and 92.1 g/100g for crude protein, 1.3 and 0.5 g/100g for crude fat, respectively). This difference is reflected in the amount of protein, corn starch, and soybean oil sources in each diet. We added this explanation under Table 3 (p. 5, L254-257).
(21) Lines 257-283-307-320: The description of the control diet does not seem adequate. The FO diet also contains casein and all three diets contain soyabean oil. In materials and methods, it has already been described, in my opinion it is not necessary to indicate anything.
Response:
The instances mentioned have been deleted accordingly.
(22) Line 318: …“compared to casein and soybean oil”… should be changed by “control diet for example”.
Response:
The sentence was modified accordingly (p. 11, L701).

Round 2
Reviewer 2 Report
I accept the manuscript in the present form after the modifications included and the replies and explanations provided.
I apologies for an alphabetical mistake in one of my comments. I wrote FO instead FP but I was certainly referring to FP (Comment number 15: ...“Future studies should be done to prove the tendency found to improve short-term memory of FO in this study”). My comment is not a misinterpretation of the results it was a written mistake. In any case, I agree with the modification included.